# MotifGrIm: Motif-Based Multi-Granularity Graph-Image Pretraining for Molecular Representation Learning

## Abstract

Molecular representation learning is widely considered as a crucial task in computer-aided molecular applications and design. Recently, many studies have explored pretraining models on unlabeled data to learn molecular structures and enhance the performance of downstream tasks. However, existing methods mainly focus on graph domains, with limited attention to other modals, such as the images. In addition, most existing methods focus on the atomic or molecular level, which leads to the neglect of high-order connection information or local structure information. In this work, we propose a motif-based multi-granularity graph-image pretraining framework, *MotifGrIm*, for molecular representation learning. In this framework, we incorporate motifs into the image domain for the first time, by generating distinct background features for different motifs in molecular images, offering a novel approach to enhancing molecular representation. Through contrastive learning within and across modules, we effectively tackle two key challenges in molecular motif pretraining with graph neural networks: (1) the over-smoothing problem, which restricts GNNs to shallow layers and hinders global molecular information capture, and (2) the aggregation of motif nodes, which leads to the loss of connectivity information between motifs. Additionally, to more effectively capture information across different molecular granularities, we propose a multi-granularity prediction pretraining strategy to optimize the model. For downstream tasks, we use only the graph encoders for prediction, reducing both time and memory consumption. We evaluate MotifGrIm on molecular property prediction and long-range benchmarks. Across eight commonly used molecular property prediction datasets, MotifGrIm outperforms state-of-the-art models with an average ROC-AUC improvement of 1.16% and achieves the best results on five of them. On long-range datasets, MotifGrIm improves the performance by at least 14.8%.

## 1 Introduction

Molecular representation learning is crucial in many fields, such as cheminformatics Aldeghi & Coley (2022); Nguyen-Vo et al. (2024), drug design Fang et al. (2022); Xue et al. (2020), and materials science Damewood et al. (2023); Fuhr & Sumpter (2022). By converting molecular structures and properties into embeddings suitable for machine learning, it enables many important tasks, such as molecular property prediction Sultan et al. (2024); Zhang et al. (2024), molecular activity evaluation Hadiby & Ben Ali (2024); Li & Fourches (2020), and molecular generation Bagal et al. (2021); Hua et al. (2024). Traditional methods often rely on manually designed molecular fingerprints (e.g., ECFP Rogers & Hahn (2010) and MACCS Durant et al. (2002)) , requiring chemists to define specific professional rules. It is time-consuming and exhibits limitations in capturing the latent complex patterns in molecules.

Driven by the success of the pretraining-finetuning paradigm, unsupervised pretrained molecular graph neural networks have achieved remarkable results in downstream tasks and are becoming increasingly popular. The key to pretraining lies in designing an effective proxy task (i.e., training objective) to leverage large-scale unlabeled datasets for optimization. Current optimization methods fall into three categories, namely, predictive methods, generative methods and contrastive learning

methods. Predictive methodsHu et al. (2019); Rong et al. (2020) use large-scale unlabeled datasets to generate chemical and topological labels for training. Despite their simplicity, these methods are susceptible to negative transfer. Generative methodsHu et al. (2020b); Zhang et al. (2021) aim to learn the distribution of molecular graphs, while contrastive learning methodsWang et al. (2022); Stärk et al. (2022); Xiang et al. (2023); Luong & Singh (2024) focus on learning a robust embedding space by aligning different molecular views.

Recently, many studies have employed multi-modal contrastive learning to align semantic information, transferring chemical knowledge from other domains, such as molecular SMILES, 3D molecular structures, and molecular images to 2D molecular graphs, thereby enhancing molecular representation. GraphMVPLiu et al. (2021a) performs self-supervised learning by leveraging the correspondence and consistency between 2D topological structures and 3D geometric views, allowing the pretrained graph encoder to implicitly encode such information. MoleculeSTMLiu et al. (2023) works by learning molecular chemical structures alongside their textual descriptions, while CGIPXiang et al. (2023) works by learning explicit information from graphs and implicit information from images in large-scale unlabeled molecular datasets. Furthermore, SGGRLWang et al. (2024) integrates the sequential, graphical, and geometric features of molecules to provide a more comprehensive representation of the molecular structures.

Existing multi-modal contrastive learning methods primarily focus on molecular-level comparisons, often overlooking finer details. However, the motif information, such as functional groups and ring structures, is critical for molecular analysis. Recent studiesZhang et al. (2020; 2021); Luong & Singh (2024) have introduced motif-based pretraining methods to capture local substructural information within graphs, demonstrating the effectiveness of motif information. However, these approaches remain graph-based and have not been extended to incorporate information from other domains.

In this work, we propose a **motif**-based multi-granularity **gr**aph-**im**age pretraining method (**MotifGrIm**) for molecular representation learning, augmenting motif data by incorporating the image modality in addition to graphs. For each molecule, we obtain four representations, namely, molecular graph, motif graph, molecular image, and motif image. Notably, this is the first application of motifs in the image domain. By coloring all atoms and bonds belonging to the same motif and adding a background feature to each motif, the model can recognize it as a substructure when processing the image. This approach enhances the capture of molecular substructure information. These inputs are then processed by two graph encoders and one image encoder, capturing molecule-level information related to global structures and motif-level information concerning substructures and higher-order connectivity from both graphical and image data.

By integrating the image module with a convolutional neural network (CNN) , we effectively address challenges at both the molecular and motif levels: (1) At the molecular level, shallow GNNs struggle to capture overall molecular structure. For instance, a two-layer GNN may fail to recognize ring structures, while deeper GNNs are often prone to oversmoothing. In contrast, the image module leverages CNNs to easily capture global structures in images. (2) At the motif level, motif graphs are constructed by aggregating nodes, which often results in the loss of internal connectivity information within the molecule. However, motif images not only preserve the specific internal structures of the entire motif but also reveal how motifs are connected through specific atoms.

Additionally, to better capture information across multiple granularities, we propose three predictive pretraining tasks. (1) Given a molecular graph embedding, predict the motifs present in the molecule; (2) Given the node embeddings of a motif graph, predict the structures it contains(e.g., atoms, heteroatoms, ions) . (3) Given a motif image, predict the shape of its corresponding motif structure graph. These tasks assist the model in achieving a deeper structural understanding and can be performed alongside other pretraining tasks.

Our contribution can be summarised as follows:

- We propose a motif-based multi-modal self-supervised learning framework for molecular representation. To the best of our knowledge, this is the first approach to extend motif knowledge from the graph domain to the multi-modal domain.

- We propose MotifGrIm, a motif-based multi-granularity graph-image pretraining method that leverages contrastive learning at both the molecular and motif levels to integrate information from graph and image data, enabling a comprehensive capture of molecular structural features. Additionally, we

introduce three predictive learning tasks at different granularities to enhance the model's structural understanding.

- Through extensive experiments on molecular property prediction and long-range benchmarks, we demonstrate the superiority and generalizability of our method. Across eight molecular property prediction datasets, our approach achieves an average AUC-ROC improvement of 1.16%, with five datasets attaining the best performance. On the long-range datasets, performance improves by at least 14.8%.

## 2 METHODOLOGY

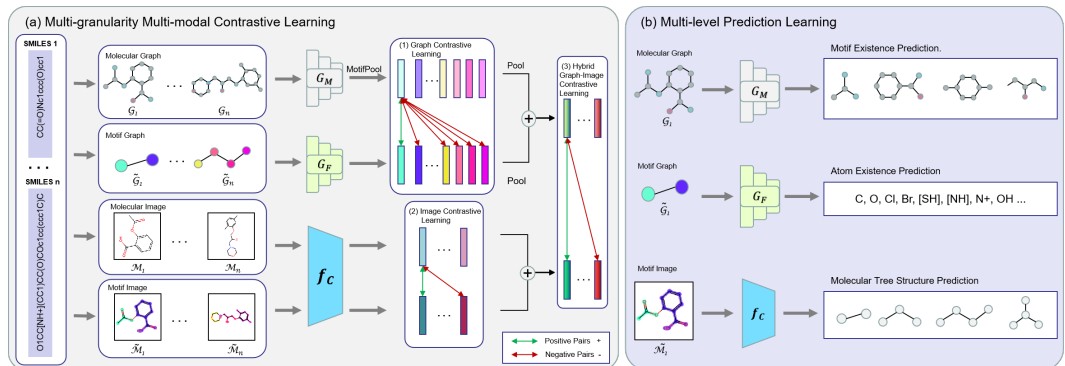

Figure 1: The overall framework of MotifGrIm. Figure (a) illustrates the Motif-based Multi-modal Contrastive Learning. The left part shows four views of the molecule generated by smiles according to Section 2.1. The (1) , (2) , and (3) respectively show the graph contrastive learning , the image contrastive learning , and the hybrid graph-image contrastive learning. Figure (b) illustrates the multi-granularity prediction learning, where the molecular graph is used for motif existence prediction, the motif graph is used for atom existence prediction, and the motif image is used for molecular tree structure prediction.

In this section, we present the motif-based multi-granularity graph-image learning framework to learn molecular representations without supervision. We first introduce how to obtain data of different levels from molecular SMILES. Then, we present the pretraining strategies of the model, including motif-based multi-modal contrastive learning and multi-granularity prediction learning. The overview of the proposed MotifGrIm is shown in Figure1.

### 2.1 DATA PREPARATION

Typically, the initial data for molecules are in the form of SMILES and we first convert the molecular SMILES notation $X_{\mathcal{S}} = \{\mathcal{S}_1, \mathcal{S}_2, \cdots, \mathcal{S}_n\}$ into two different modes: molecular images $X_{\mathcal{M}} = \{\mathcal{M}_1, \mathcal{M}_2, \cdots, \mathcal{M}_n\} \in \mathbb{R}^{224 \times 224 \times 3}$ and molecular graphs $X_{\mathcal{G}} = \{\mathcal{G}_1, \mathcal{G}_2, \cdots, \mathcal{G}_n\}$, where $n$ is the total number of molecules. The molecular graphs $\mathcal{G}$ can be represented as $\mathcal{G} = (\mathcal{V}, \mathcal{E})$, where vertices $\mathcal{V}$ and edges $\mathcal{E}$ represent the sets of atoms and bounds, respectively.

#### 2.1.1 MOTIF SEGMENTATION METHODS

Inspired by chemistry, existing methods such as BRICSDegen et al. (2008) and RECAPLewell et al. (1998) rely on rule-based program variants. However, the extracted vocabulary is often quite large and contains many specific or low-frequency fragments which pose significant challenges for motif recognition. To overcome these shortcomings, previous work has proposed further decomposing fragmentsZhang et al. (2021), even down to the level of rings and bondsJin et al. (2018), but the reduction in fragment size is detrimental to the ability to capture the higher-order representations of molecules. To alleviate the above issues, we draw on the Principal Subgraph Mining algorithm from Kong et al. (2022) in this work.

### 2.1.2 Subgraph Mining Algorithm

For a graph $\mathcal{G} = \{\mathcal{V}, \mathcal{E}\}$, we define a subgraph of $\mathcal{G}$ as $\mathcal{S} = (\tilde{\mathcal{V}}, \tilde{\mathcal{E}})$, where $\tilde{\mathcal{V}} \subseteq \mathcal{V}$ and $\tilde{\mathcal{E}} \subseteq \mathcal{V}$. We call a subgraph $\mathcal{S}$ a principal subgraph if, for any other subgraph $\mathcal{S}'$ that spatially intersects with $\mathcal{S}$ in a molecule, it holds that either $\mathcal{S}' \subseteq \mathcal{S}$ or $c(\mathcal{S}') \leq c(\mathcal{S})$, where $c(\mathcal{S})$ is the frequency of subgraph $\mathcal{S}$ among molecules. Intuitively, principal subgraphs are fragments that are both larger in size and occur more frequently. The algorithm heuristically builds a vocabulary of these principal subgraphs through the following steps:

**Initialization.** Initialize the vocabulary $\mathbb{V}$ with all unique atoms.

**Merge.** For each molecular graph, for all pairs of overlapping fragments, merge the fragments in the pair and update the occurrence counts of the newly formed combined fragment.

**Update.** We count the frequency of each merged subgraph appears in the last stage. We select the most frequent one as a new fragment in the vocabulary $\mathbb{V}$. Then, we go back to the merge stage until the vocabulary size reaches the set number $N$.

For more details about the algorithm, please refer to Kong et al. (2022).

### 2.1.3 Motif Dataset Construction

After performing Subgraph Mining Algorithm on the molecules, we use the extracted principal vocabulary $\mathbb{V}$ to construct the motif dataset.

**Motif Graph.** For a molecular graph $\mathcal{G} = (\mathcal{V}, \mathcal{E})$, let $F = \{S^{(0)}, S^{(1)}, \cdots, S^{(m)}\}$ be its corresponding motif, where $S^{(i)} = (\tilde{\mathcal{V}}^{(i)}, \tilde{\mathcal{E}}^{(i)}) \in \mathbb{V}$ is a subgraph, $\tilde{\mathcal{V}}^{(i)} \cap \tilde{\mathcal{V}}^{(j)} = \emptyset$ and $\cup_{i=1}^{m} \tilde{\mathcal{V}}^{(i)} = \mathcal{V}$. We denote the motif graph of $\mathcal{G}$ as $\tilde{\mathcal{G}} = (\tilde{\mathcal{V}}, \tilde{\mathcal{E}})$, where $|\tilde{\mathcal{V}}| = |F|$ and each node $v_F^{(i)} \in \tilde{\mathcal{V}}$ corresponds to a motif $S^{(i)}$. If there is at least one edge between the atoms of two motifs, then there is an edge between the nodes of these two motifs, defined as $\tilde{\mathcal{E}} = \{(i, j) | \exists u, v, u \in \tilde{\mathcal{V}}^{(i)}, v \in \tilde{\mathcal{V}}^{(j)}, (u, v) \in \mathcal{E}\}$. For simplicity, we retained the edge features within each motif, and the node features of the motifs are derived from embeddings in an optimizable lookup table.As a result, we obtained the motif graph dataset $\tilde{X}_{\mathcal{G}} = \{\tilde{\mathcal{G}}_1, \tilde{\mathcal{G}}_2, \cdots, \tilde{\mathcal{G}}_n\}$.

**Motif Image.** To enable the image encoder model to better recognize the motif structures of molecules, we innovatively propose constructing motif images for training. Specifically, we first initialize a color matrix $\mathbb{C} \in \mathbb{R}^{K \times 3}$, where the differences between each color are maximized to ensure the greatest possible contrast and K is the number of color. For motif $S^{(i)} \in \mathbb{V}$, we assign corresponding color is $\mathbb{C}[r(S^{(i)}) \% K]$, where $r(S^{(i)})$ represents the result of sorting motif $S^{(i)}$ first by size and then by frequency. Then, we determine which motif each atom and bond in the molecule belongs to. Finally, we apply corresponding background colors to the atoms and bonds of the molecule, resulting in the motif image dataset $\tilde{X}_{\mathcal{M}} = \{\tilde{\mathcal{M}}_1, \tilde{\mathcal{M}}_2, \cdots, \tilde{\mathcal{M}}_n\} \in \mathbb{R}^{224 \times 224 \times 3}$. In summary, in each molecular image, we add a background with different colors for the nodes and edges corresponding to the different motif structures of the molecule. This provides each motif with a new image feature, allowing the image encoder to better capture higher-order structural patterns.

### 2.2 Motif-based Multi-modal Contrastive Learning

In this section, we perform motif-based multi-modal contrastive learning using graph and image encoders based on the data obtained in Section 2.1. The contrastive framework is illustrated in Figure 1(a) .

### 2.2.1 Graph Contrastive Learning

For graph data, we define two graph encoders, $G_M$ and $G_F$ , to encoder the molecular graphs and motif graphs, respectively. Given a molecular graph $\mathcal{G}_i = (\mathcal{V}_i, \mathcal{E}_i)$ and the corresponding motif graph $\tilde{\mathcal{G}}_i = (\tilde{\mathcal{V}}_i, \tilde{\mathcal{E}}_i)$, we can obtain their final node embeddings $H^{(i)}$ and $\tilde{H}^{(i)}$ as follows:

$$H_N^{(i)} = G_M(\mathcal{V}_i, \mathcal{E}_i), \tilde{H}_F^{(i)} = G_F(\tilde{\mathcal{V}}_i, \tilde{\mathcal{E}}_i) \tag{1}$$

Considering that motif-level data are beneficial for a deeper understanding of molecular properties, we incorporate motif-level representations in our graph contrastive learning. Since each row of $\tilde{H}^{(i)}$ represents the embedding of a motif, we need to apply pooling to the data in $H^{(i)}$ before performing

contrastive learning while each row of $H^{(i)}$ represents the embedding of an atom. We first define a function MOTIFPOOL which performs an average pooling operation on the atom embeddings that belong to the same motif:

$$H_F^{(i)} = \text{MOTIFPOOL}(H_N^{(i)}, \text{MAP}(\mathcal{V}_i, \tilde{\mathcal{V}}_i)) \tag{2}$$

where $\text{MAP}(\cdot)$ is mapping function used to establish the relationship between motifs and atoms. Then we minimize the graph contrastive learning objective based on the InfoNCE loss:

$$\mathcal{L}_{\mathcal{G}} = -\log \frac{exp\left(\left\langle H_{F,r}^{(i)}, \tilde{H}_{F,r}^{(i)} \right\rangle\right)}{\sum_{j=1}^{n} \sum_{k=1}^{|\tilde{\mathcal{V}}_j|} exp\left(\left\langle H_{F,r}^{(i)}, \tilde{H}_{F,k}^{(j)} \right\rangle\right)} \tag{3}$$

where $H_{F,r}^{(i)}$ and $\tilde{H}_{F,r}^{(i)}$ are the $r$-th row of $H_F^{(i)}$ and $\tilde{H}_F^{(i)}$, respectively. $|\tilde{\mathcal{V}}_j|$ is the number of motifs in the $j$-th molecule.

### 2.2.2 IMAGE CONTRASTIVE LEARNING

Unlike previous methods that applied different data augmentations to molecular images for contrastive learning, we innovatively incorporate motif-level data in molecular images. By marking motifs with different colors, we set to generate motif-based molecular images and use contrastive learning to optimize the model.

We first use an image encoder $f_C$ to process both molecular images and motif images. For a molecular image $\mathcal{M}_i$ and its corresponding motif image $\tilde{\mathcal{M}}_i$, their final embeddings $h_m^{(i)}$ and $\tilde{h}_m^{(i)}$ are produced respectively:

$$H_m^{(i)} = f_C(\mathcal{M}_i), \tilde{H}_m^{(i)} = f_C(\tilde{\mathcal{M}}_i) \tag{4}$$

Similarly, we optimize the image contrastive learning process using the InfoNCE loss function:

$$\mathcal{L}_{\mathcal{M}} = -\log \frac{exp\left(\left\langle H_m^{(i)}, \tilde{H}_m^{(i)} \right\rangle\right)}{\sum_{j=1}^{n} exp\left(\left\langle H_m^{(i)}, \tilde{H}_m^{(j)} \right\rangle\right)} \tag{5}$$

By bringing two image views of the same molecule as close as possible in the feature space while pushing apart images of different molecules, an effective molecular image representation can be learned.

### 2.2.3 HYBRID GRAPH-IMAGE CONTRASTIVE LEARNING

To achieve compatibility and consistency between different modalities and facilitating information exchange between them, we set contrastive learning between the graph and image modalities. Unlike previous multi-modal contrastive learning methods in the molecular field, which only performed contrastive learning at the molecular level, we propose a hybrid graph-image contrastive learning method.
We first obtain the final embeddings $H_g^{(i)}$ and $\tilde{H}_g^{(i)}$ for the molecular graph and motif graph through node aggregation:

$$H_g^{(i)} = \text{POOL}(H_F^{(i)}), \tilde{H}_g^{(i)} = \text{POOL}(\tilde{H}_F^{(i)}) \tag{6}$$

where $\text{POOL}(\cdot)$ is average pooling function.
This method utilizes both molecular-level and motif-level data, with its corresponding InfoNCE loss function as:

$$\mathcal{L}_{\mathcal{G}\mathcal{M}} = -\log \frac{exp\left(\left\langle S_g^{(i)}, S_m^{(i)} \right\rangle\right)}{\sum_{j=1}^{n} exp\left(\left\langle S_g^{(i)}, S_m^{(j)} \right\rangle\right)} \tag{7}$$

$$S_g^{(i)} = \beta H_g^{(i)} + (1-\beta)\tilde{H}_g^{(i)}, S_m^{(i)} = \beta H_m^{(i)} + (1-\beta)\tilde{H}_m^{(i)} \tag{8}$$

where $\beta$ is the weight hyperparameter.

### 2.3 MULTI-GRANULARITY PREDICTION LEARNING

To enable the model to capture information at multiple granularities, we propose three predictive learning tasks to optimize the three encoders. The prediction framework is shown in Figure1(b) .

### 2.3.1 Atom Existence Prediction

In the motif graphs, we design an atom existence prediction method to enhance the representation of atomic information by the motif graph encoder $G_F$. We perform multi-label prediction for each motif to indicate which atoms are included in that motif. Notably, to better represent atomic information, we collected some frequently occurring heteroatoms and ions for prediction, such as [SH] , [NH] , N+, etc. Assuming the set of atoms is $|\mathbb{A}|$, and the atomic label of motif $i$ is $y_a^{(i)} \in \{0, 1\}^{|\mathbb{A}|}$, the loss for atom existence prediction is:

$$\mathcal{L}_{\mathcal{A}} = \frac{1}{|\mathbb{A}|} \sum_{j=1}^{|\mathbb{A}|} \left| [y_a^{(i)}]_j - [f_a(\tilde{H}_{F,r}^{(i)})]_j \right| \tag{9}$$

where $f_a$ is an MLP used for atom existence prediction and $[\cdot]_j$ is the $j$-th element of the vector.

### 2.3.2 Motif Existence Prediction

To enable the molecular graph encoder $G_M$ to better represent the structure of motifs, we designed a motif existence prediction method to identify the types of motifs contained within each molecule. Assuming the motif label of the molecule $i$ is $y_m^{(i)} \in \{0, 1\}^{|\mathbb{V}|}$, where $|\mathbb{V}|$ is the number of motifs, the loss for motif existence prediction is:

$$\mathcal{L}_{\mathcal{F}} = \frac{1}{|\mathbb{V}|} \sum_{j=1}^{|\mathbb{V}|} \left| [y_m^{(i)}]_j - [f_f(H_g^{(i)})]_j \right| \tag{10}$$

where $f_f$ is an MLP used for motif existence prediction.

### 2.3.3 Molecular Tree Structure Prediction

In the motif images, we use different colors to label the motifs to help the image encoder recognize the motifs of the molecules. Furthermore, to enable the image encoder $f_C$ to effectively identify the connecting structures between motifs, we propose a molecular structure prediction method to identify the structural backbone of the motif images which are represented in a tree structure formed by motifs. If two molecules share the same structural backbone, they will yield the same prediction result. For instance, if a molecule consists of four motifs arranged in a straight line, its corresponding tree structure will have a width of 1 and a depth of 4. Thanks to the sufficiently large motifs generated by the subgraph mining algorithm and the relatively small number of motifs in the molecule, we can enumerate all possible tree structures. Assuming the structural tree set is $\mathbb{T}$ and the structural label of the molecule $i$ is $y_t^{(i)} \in \mathbb{R}^{|\mathbb{T}|}$, the molecular structure prediction loss function is:

$$\mathcal{L}_{\mathcal{T}} = -y_t^{(i)} \log(f_t(\tilde{H}_m^{(i)})) \tag{11}$$

where $f_t$ is an MLP used for molecular tree structure prediction.

Finally, we combine the contrastive loss $\mathcal{L}_{\mathcal{C}}$ from Section 2.2 and the prediction loss $\mathcal{L}_{\mathcal{P}}$ from Section 2.3, with $0 < \alpha < 1$ as the weight hyperparameter, to form the overall loss function $\mathcal{L}$ of the model as follows:

$$\mathcal{L} = \alpha \mathcal{L}_{\mathcal{C}} + (1 - \alpha) \mathcal{L}_{\mathcal{P}} \tag{12}$$
$$\mathcal{L}_{\mathcal{C}} = \mathcal{L}_{\mathcal{G}} + \mathcal{L}_{\mathcal{M}} + \mathcal{L}_{\mathcal{GM}} \tag{13}$$
$$\mathcal{L}_{\mathcal{P}} = \mathcal{L}_{\mathcal{T}} + \mathcal{L}_{\mathcal{F}} + \mathcal{L}_{\mathcal{A}} \tag{14}$$

After pretraining, three models are produced. Since generating image data is information-intensive and resource-consuming, to save training time and resources while allowing the model to generalize to more domains, we propose using only the embeddings generated by the two graph encoders $G_M$ and $G_F$, concatenating the embeddings for downstream tasks.

## 3 Experiments

### 3.1 Experimental settings

**Datasets.** During pretraining, we utilized 456K unique unlabeled molecular SMILES strings from the ChEMBL database. We selected a motif vocabulary with a size of 800. In the fine-tuning

stage, we conducted experiments on 10 downstream benchmarks, including 8 molecular property prediction datasets (BBBP, Tox21, ToxCast, SIDER, ClinTox, BACE, HIV, and MUV) and molecular prediction tasks on two large peptide datasets from the Long-range Graph Benchmark, namely, Peptide-Func (graph classification) and Peptide-Struct(graph regression) Dwivedi et al. (2022). We used the RdkitLandrum et al. (2013) library to generate molecular images based on SMILES strings, and utilized the OGBHu et al. (2020a) library to abstract the SMILES strings into graphs, initializing the node and edge features within the graphs.

**Implementation details.** According to the setting of the previous method, we model our molecular graph encoders $G_M$ with 5-layer GIN and motif graph encoder $G_F$ with 2-layer GIN in molecular classification benchmarks. For the long-range benchmarks, both $G_M$ and $G_F$ utilize a 5-layer GIN architecture, with the molecular graph featurization derived from the Open Graph Benchmark. For the image encoder, we selected the Swin-BLiu et al. (2021b) pretrained on ImageNet-1K to extract features from molecular images. We fixed the shallower layers parameters while fine-tuning the the deeper one during pretraining. In the pretraining stage, the graph encoders and the image encoder receive graphs and images, respectively, and embed the multi-modal data into a shared feature space of 300 dimensions. We use the Adam optimizer to train ours framework by setting the initial learning rate to 0.001, the number of epoches to 100, and the batch size to 256. We reduce the learning rate by a factor of 0.1 every 5 epochs without an improvement. In the fine-tuning stage, we concatenate the embeddings generated by the graph encoders and add a MLP to make predictions for the downstream tasks. More implementation details are show in Appendix C.

**Baselines.** As shown in Table 1, we compare our model with several notable pretrained baselines in the molecular property prediction task, including predictive methods (AttrMask & ContextPredHu et al. (2019), G-Motif & G-Contextual Rong et al. (2020)), generative method (GPTGNNHu et al. (2020b)), graph contrastive methods (GraphLoG Xu et al. (2021), GraphCL You et al. (2020), JOAO, JOAOv2You et al. (2021)), motif-based method(MGSSLZhang et al. (2021), GraphFPLuong & Singh (2024), DGPMYan et al. (2024)) and multi-modal contrastive methods(GraphMVPLiu et al. (2021a), CGIPXiang et al. (2023), MoleculeSTMLiu et al. (2023)). For long-range prediction, we compare our method with popular GNN architectures: GCN Kipf & Welling (2016), GCNII Chen et al. (2020), GINXu et al. (2018), and GatedGCN Bresson & Laurent (2017) with and without Random Walk Spatial Encoding Dwivedi et al. (2021).

## 3.2 RESULT ON MOLECULAR PROPERTY PREDICTION BENCHMARK

Table 1 report the result on 8 molecular property prediction benchmark. Except for CGIPXiang et al. (2023), the results of other baselines are collected from the literatureXu et al. (2021); You et al. (2021; 2020); Luong & Singh (2024); Yan et al. (2024); Liu et al. (2023). To ensure a comprehensive evaluation, we conducted experiments on several variants of motifGrIM, including (1) without the image, multi-granularity prediction learning and hybrid graph-image contrastive learning module(w.o. IPH) , (2) without the image and hybrid graph-image contrastive learning module(w.o. IH) , (3) without the motif-based contrastive learning and hybrid graph-image contrastive learning module(w.o. CH) , (4) without the multi-granularity prediction learning module(w.o. P) , and (5) without the hybrid graph-image contrastive learning(w.o. H) . For the variants that include contrastive learning and prediction learning module, we select $\alpha = 0.9$. As for variants equipped with an image module, we opt for $\beta = 0.7$. The experimental results are shown in Table 1, where we summarize all methods into six different categories.

From the results, we can see that compared with previous studies, our approach has shown significant improvement. Among the 8 datasets, we achieved the best performance on 5 of them. Furthermore, our proposed method outperforms others in both average ranking and average AUC. The average AUC of ours MotifGrIm method has improved by 1.15% compared to the current best-performing model, MoleculeSTM. Furthermore, by comparing MotifGrIM with its five variants, it is clear that as more modules are added, the model's performance gradually improves, with MotifGrIM outperforming its five variants. This demonstrates the importance of each module to the MotifGrIM. The variant without the multi-granularity prediction learning module performs worse than the one without the hybrid contrastive learning module, indicating that the prediction learning module is more crucial. And variants without the image module show a significant drop in performance, highlighting the necessity and effectiveness of adding the image module for model pretraining. Finally, by summarizing the methods into 6 different categories, we observe that MotifGrIM and its variants

Table 1: Test ROC-AUC on eight molecular property prediction datasets. The mean and standard deviation are reported for five random seeds. We summarize the methods into 6 categories: (1) No-pretrain; (2) prediction and generation methodsHu et al. (2019); Rong et al. (2020); Hu et al. (2020b); (3) graph contrastive methodsXu et al. (2021); You et al. (2020; 2021); (4) motif-based methodsZhang et al. (2021); Yan et al. (2024); Luong & Singh (2024); (5) multi-modal contrastive methodsLiu et al. (2021a); Xiang et al. (2023); Liu et al. (2023); (6) MotifGrIm and its five variants. **Bold** indicates the best performance and underline indicates the second best one.

| Method | BBBP | Tox21 | Toxcast | SIDER | ClinTox | MUV | HIV | BACE | Avg.Rank | Avg.AUC |
|---|---|---|---|---|---|---|---|---|---|---|
| No-pretrain | 65.6±1.4 | 71.5±1.0 | 61.5±0.8 | 59.4±1.2 | 66.5±5.2 | 74.5±0.5 | 64.4±1.9 | 72.6±1.9 | 19.50 | 67.00 |
| AttrMaskingHu et al. (2019) | 64.3±2.8 | 76.7±0.4 | 64.2±0.5 | 61.0±0.7 | 71.8±4.1 | 74.7±1.4 | 77.2±1.1 | 79.3±1.6 | 10.50 | 71.15 |
| ContextPredHu et al. (2019) | 68.0±2.0 | 75.7±0.7 | 63.9±0.6 | 60.9±0.6 | 65.9±3.8 | 75.8±1.7 | 77.3±1.0 | 79.6±1.2 | 9.63 | 70.89 |
| G-MotifRong et al. (2020) | 66.9±3.1 | 73.6±0.7 | 62.3±0.6 | 61.0±1.5 | 77.7±2.7 | 73.0±1.8 | 73.8±1.2 | 73.0±3.3 | 17.38 | 70.16 |
| G-ContextualRong et al. (2020) | 69.9±2.1 | 75.0±0.6 | 62.8±0.7 | 58.7±1.0 | 60.6±5.2 | 72.1±0.7 | 76.3±1.5 | 79.3±1.1 | 14.63 | 69.34 |
| GPT-GNNHu et al. (2020b) | 64.5±1.4 | 74.9±0.3 | 62.5±0.4 | 58.1±0.3 | 58.3±5.2 | 75.9±2.3 | 65.2±2.1 | 77.9±3.2 | 16.00 | 67.16 |
| GraphLoGXu et al. (2021) | 67.8±1.9 | 75.1±1.0 | 62.4±0.2 | 59.5±1.5 | 65.3±3.2 | 73.6±1.2 | 73.7±0.9 | 80.2±3.5 | 15.25 | 69.70 |
| GraphCLYou et al. (2020) | 69.7±0.7 | 73.9±0.7 | 62.4±0.6 | 60.5±0.9 | 76.0±2.7 | 69.8±2.7 | **78.5±1.2** | 75.4±1.4 | 14.88 | 70.78 |
| JOAOYou et al. (2021) | 70.2±1.0 | 75.0±0.3 | 62.9±0.5 | 60.0±0.8 | 81.3±2.5 | 71.7±1.4 | 76.7±1.2 | 77.3±0.5 | 12.50 | 71.89 |
| JOAOv2You et al. (2021) | 71.4±0.9 | 74.3±0.6 | 63.2±0.5 | 60.5±0.7 | 81.0±1.6 | 73.7±1.0 | 77.5±1.2 | 75.5±1.3 | 11.75 | 72.14 |
| MGSSLZhang et al. (2021) | 68.9±2.5 | 74.9±0.6 | 63.3±0.5 | 57.7±0.7 | 67.5±5.5 | 73.2±1.9 | 75.7±1.3 | 82.1±2.7 | 14.00 | 70.41 |
| GraphFPLuong & Singh (2024) | 72.0±1.7 | 74.0±0.7 | 65.3±0.5 | 63.6±1.2 | 84.7±5.8 | 75.4±1.9 | 78.0±1.5 | 80.5±1.8 | 7.25 | 74.01 |
| DGPMYan et al. (2024) | 71.2±0.5 | 75.3±0.4 | 64.0±0.7 | 60.3±0.8 | 80.9±1.3 | 75.3±1.6 | 77.3±0.6 | 81.1±0.7 | 7.38 | 73.17 |
| GraphMVPLiu et al. (2021a) | 68.5±0.2 | 74.5±0.4 | 62.7±0.1 | 62.3±1.6 | 79.0±2.5 | 75.0±1.4 | 74.8±1.4 | 76.8±1.1 | 13.38 | 71.70 |
| CGIPXiang et al. (2023) | 66.9±1.0 | 71.5±0.5 | 63.5±0.2 | 58.4±0.6 | 88.0±1.5 | 75.3±1.2 | 78.2±0.6 | 76.4±0.7 | 12.13 | 72.64 |
| MoleculeSTMLiu et al. (2023) | 70.1±0.5 | **76.9±0.5** | 65.1±0.4 | 61.0±1.1 | **92.5±1.1** | 73.4±2.0 | 76.9±1.8 | 80.8±1.3 | 7.00 | 74.57 |
| MotifGrIm(w.o. ICH) | 67.5±2.3 | 73.2±0.8 | 62.8±0.6 | 64.1±1.4 | 73.6±2.4 | 74.3±1.1 | 73.2±3.2 | 76.4±1.7 | 15.25 | 70.64 |
| MotifGrIm(w.o. IPH) | 70.1±1.8 | 74.3±0.3 | 63.9±0.8 | 63.6±1.0 | 87.7±5.8 | 74.5±1.8 | 76.1±2.0 | 77.1±2.1 | 10.75 | 73.73 |
| MotifGrIm(w.o. IH) | 69.7±2.3 | 74.8±0.6 | 64.8±0.4 | **65.2±1.5** | 88.8±1.9 | 74.6±1.4 | 77.2±1.1 | 77.3±1.6 | 8.25 | 74.05 |
| MotifGrIm(w.o. P) | 70.5±0.8 | 74.8±0.5 | 64.3±0.2 | 63.8±1.2 | 89.6±1.6 | 76.2±0.9 | 76.6±1.4 | **83.4±1.7** | 5.88 | 74.90 |
| MotifGrIm(w.o. H) | **72.6±1.2** | 75.8±0.4 | 65.0±0.6 | 63.6±1.0 | 90.5±2.1 | 75.5±1.2 | 77.3±1.5 | 82.4±0.8 | 3.63 | 75.34 |
| MotifGrIm | 70.7±1.4 | 75.4±0.8 | **65.4±0.7** | 64.2±0.8 | 91.2±1.6 | **78.4±1.1** | 77.8±1.0 | 82.7±1.6 | 2.75 | **75.73** |

Table 2: The impact of pretraining with different image encoders on downstream performance.

| Image Encoder | BBBP | Tox21 | Toxcast | SIDER | ClinTox | MUV | HIV | BACE | Avg.Rank | Avg.AUC |
|---|---|---|---|---|---|---|---|---|---|---|
| ResNet18 | 70.9 | 74.9 | 65.1 | 63.9 | 89.6 | 77.2 | **78.4** | 81.6 | 3.25 | 75.20 |
| ResNet50 | **71.7** | 74.4 | 64.3 | 64.4 | **91.4** | 76.8 | 78.1 | 82.2 | 2.75 | 75.41 |
| Swin-T | 70.5 | 74.8 | 65.0 | 64.4 | 90.1 | 76.4 | 78.1 | 82.1 | 3.63 | 75.18 |
| Swin-S | 71.2 | **75.5** | 64.8 | **64.5** | 90.8 | 76.5 | 78.2 | 81.9 | 2.63 | 75.43 |
| Swin-B | 70.7 | 75.4 | **65.4** | 64.2 | 91.2 | **78.4** | 77.8 | **82.7** | 2.50 | **75.73** |

have the best average ROC-AUC, while no-pretrain is the worst. Among the remaining pretraining methods, the prediction and generation methods perform the worst, followed by the contrastive learning methods, while multi-modal contrastive method and the motif-based method show similar and optimal performance. It can see that the motif-based method and multi-modal contrastive learning method are the most effective existing methods for improving model performance. And our proposed MotifGrIm innovatively extends motifs from the graph domain to the image domain, combining both methods for the first time, thus achieving the best performance.

The experimental results on long-range benchmark is provided in Appendix D.

## 3.3 Pretraining with Various Image Encoders

To examine the impacts of different image encoders on the model performance, we select five image encoders—ResNet18, ResNet50, Swin-T, Swin-S, and Swin-B—for model pretraining. All encoders were pretrained on ImageNet-1K. As shown in Table 2, we report the model performance on eight molecular property prediction datasets after pretraining with different image encoders. The experimental results indicate that Swin-B achieves the best performance. Interestingly, ResNet50 underperforms compared to ResNet18, suggesting that excessively deep image encoders may not be optimal for capturing molecular features.

## 3.4 Parameter Analysis

In order to study the impacts of different parameter selections on the final performance of the model, we set up two experiments to study the impact of weighting hyperparameter $\alpha$ and color matrix size $K$ on the model performance. For each parameter, we conducted experiments on 8 molecular property prediction datasets using 5 specified random seeds, reporting the average ROC-AUC. More parameter analysis are provided in the Appendix E.

Table 3: The impact of the weighting hyperparameter $\alpha$ used to joint the two modules.

| Method | BBBP | Tox21 | Toxcast | SIDER | ClinTox | MUV | HIV | BACE | Avg.Rank | Avg.AUC |
|---|---|---|---|---|---|---|---|---|---|---|
| MotifGrIm($\alpha$=0.5, $\beta$=0.7) | 69.5 | 74.8 | 65.1 | 64.4 | 87.6 | 75.5 | 76.6 | 82.3 | 4.50 | 74.45 |
| MotifGrIm($\alpha$=0.6, $\beta$=0.7) | 69.5 | 74.9 | 64.2 | 63.8 | 88.2 | 78.3 | 77.6 | 82.7 | 3.63 | 74.9 |
| MotifGrIm($\alpha$=0.7, $\beta$=0.7) | 69.9 | 75.2 | 63.9 | 64.1 | 90.0 | 78.1 | 77.1 | 83.1 | 3.38 | 75.18 |
| MotifGrIm($\alpha$=0.8, $\beta$=0.7) | 69.4 | **75.7** | 64.2 | **64.5** | 88.7 | 77.5 | 77.2 | 82.6 | 3.38 | 74.98 |
| MotifGrIm($\alpha$=0.9, $\beta$=0.7) | 70.7 | 75.4 | **65.4** | 64.2 | **91.2** | **78.4** | **77.8** | 82.7 | **1.63** | **75.73** |
| MotifGrIm($\alpha$=1.0, $\beta$=0.7) | 70.5 | 74.8 | 64.3 | 63.6 | 88.6 | 76.2 | 76.6 | **83.4** | 3.88 | 74.75 |

Table 4: The impact of varying color matrix sizes $K$.

| Method | BBBP | Tox21 | Toxcast | SIDER | ClinTox | MUV | HIV | BACE | Avg.Rank | Avg.AUC |
|---|---|---|---|---|---|---|---|---|---|---|
| MotifGrIm($K$=40) | 69.3 | **75.8** | 65.2 | 62.8 | 90.2 | 75.0 | 76.2 | 81.4 | 2.63 | 74.49 |
| MotifGrIm($K$=80) | **70.7** | 75.4 | **65.4** | 64.2 | **91.2** | **78.4** | **77.8** | **82.7** | **1.25** | **75.73** |
| MotifGrIm($K$=160) | 69.4 | 74.6 | **65.2** | 63.7 | 87.1 | 73.6 | 76.7 | 82.3 | 2.75 | 74.08 |
| MotifGrIm($K$=320) | 69.2 | 75.0 | 65.1 | **64.3** | 86.3 | 73.5 | 77.0 | 80.2 | 3.25 | 73.83 |

**Effect of weighting parameter** $\alpha$. In Equation equation 12, we integrate motif-based multi-modal contrastive learning and multi-granularity prediction learning by optimizing the joint objective function. To assess the impact of different $\alpha$ values, we conducted parameter analysis. We select $\alpha$ from $\{0.5, 0.6, 0.7, 0.8, 0.9, 1.0\}$ while keeping $\beta$ fixed at 0.7. Table 3 shows that the model performs the best when $\alpha = 0.9$, significantly outperforming the case of $\alpha = 1.0$, where multi-granularity prediction learning is not adopted. This result indicates that prediction learning contributes to an improvement in the model performance. Additionally, as $\alpha$ decreases, the model performance generally declines, highlighting the importance of information interaction in contrastive learning.

**Effect of color matrix size** $K$. The size $K$ of the color matrix used to generate motifs may affect the model's performance. As shown in the Table 4, the best performance is achieved when $K = 80$. Too few colors lead to many classes with the same color in the motif, while too many color matrices result in low distinguishability between different colors. Both cases make it more challenging for the model to differentiate between motifs.

### 3.5 VISUALIZING THE LEARNED EMBEDDINGS

As shown in Figure 2, we visualize the learned embeddings via t-SNE. We use $G_M$ after pretraining to generate embeddings for the motifs. Each point represents a motif, and the color of the point represents the motif category to which the point belongs. For clarity, we select the 20 frequent motif categories for data visualization. The figure shows that the model can effectively distinguish between different motif categories, indicating that the $G_M$ is capable of capturing the structural information of various motifs well.

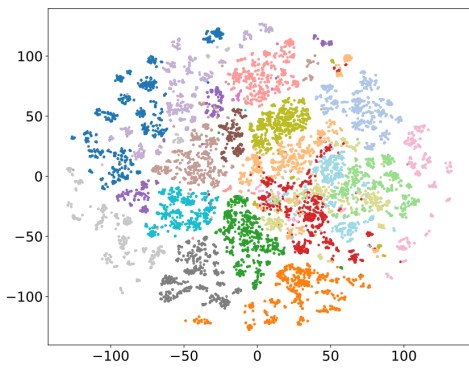

Figure 2: t-SNE plot of embeddings for different Motifs. Each color corresponds to a Motif class.

### 4 CONCLUSION AND FUTURE WORK

In this paper, we propose **Motif-GrIm**, a motif-based multi-granularity graph-image pretraining method that introduces the concept of motifs into the multi-modal domain for the first time. Using an optimized motif vocabulary, we generate data from four different perspectives and use them for pretraining with contrastive and predictive learning. Our method outperforms existing approaches, leading to superior performance in molecular property prediction and the long-range peptide dataset. While current research on molecular motifs has mainly focused on the graph domain, we extend this concept to the image domain with notable success. Moving forward, we aim to enhance molecular motif segmentation through more effective data generation methods and expand the application of motifs to other areas, such as molecular SMILES and 3D molecular structures. Additionally, we strive to develop improved molecular pretraining methods to address the challenge of limited labeled molecular data.

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

## A   APPENDIX

## B   RELATED WORK

### B.1   MOLECULAR REPRESENTATION LEARNING.

Traditional methods Carhart et al. (1985); Rogers & Hahn (2010) represent molecular structures using fingerprints. Some earlier studies Svetnik et al. (2004); Wu et al. (2018); Meyer et al. (2019) have employed tree-based machine learning models, such as Random Forest Breiman (2001) and XGBoost Chen & Guestrin (2016), to model these fingerprints for predicting molecular properties. With the advancement of neural networks, deep learning-based molecular representation learning has gradually become mainstream. String-based molecular representationsShen & Nicolaou (2019); Yüksel et al. (2023) primarily use molecular SMILES and InChI as inputs, leveraging the powerful language processing capabilities of language models to extract molecular features. 2D graph-based molecular representationsGilmer et al. (2017); Zhang et al. (2021); Chen et al. (2024) treat atoms as nodes and the bonds between atoms as edges, using graph neural networks(GNNs) to capture the internal structural information of molecules. 3D graph-based molecular representations Liu et al. (2021a); Zhou et al. further consider the 3D coordinates of atoms, which are crucial for determining certain chemical and physical properties. Molecular images provide an intuitive representation of molecular information, and recent studiesZhong et al. (2021); Zeng et al. (2022) have explored the use of molecular images to enhance the performance of molecular representation learning.

## B.2 Molecular pretraining.

Due to the sparsity of labeled molecular data and the poor generalization ability, recent studies have attempted to leverage large-scale unlabeled molecules to train models and transfer knowledge to downstream datasets with limited labels through fine-tuning. In early works, molecular pretraining primarily utilized a single modality. SMILES-BERTWang et al. (2019) employs attention-based transformer layers and is pretrained on 35 million compounds from ZINC using a masked SMILES recovery task. GROVER Rong et al. (2020) and MPG Li et al. (2021) employ a graph-based framework along with effective node and graph-level pretraining strategies to learn molecular representations. Recently, motif-based pretraining methodsZhang et al. (2020; 2021); Luong & Singh (2024) have been proposed to better extract molecular features using motifs or fragments of molecules. However, these approaches still rely on a single modality for molecular representation.

## B.3 Multi-modal molecular contrastive learning.

Recently, many studies have utilized contrastive learning on different modalities to facilitate information interaction between multiple perspectives, thereby achieving better molecular representation. MomuSu et al. (2022) pre-trains molecular representations through contrastive learning by utilizing molecular graphs and semantically related textual data, which are extracted from published scientific citation index papers. MoleculeSTMLiu et al. (2023) employs a contrastive learning strategy to jointly learn both the chemical structures of molecules and their textual descriptions. Recently, CGIPXiang et al. (2023) proposed a model that combines graph and image data using data augmentation techniques to transfer chemical knowledge from graphs to images. Git-MolLiu et al. (2024) further integrated a multimodal large language model combining graph, image, and text information to capture the rich details of molecular structures and images. However, these methods focus solely on learning molecular representations at the molecular level, overlooking the impact of motif information on molecular properties.

# C Experimental Details

## C.1 Datasets

In the fine-tuning stage, we set 10 downstream tasks, including 8 molecular property prediction datasets (BBBP, Tox21, ToxCast, SIDER, ClinTox, BACE, HIV, and MUV) and molecular prediction tasks on two large peptide datasets from the Long-range Graph Benchmark: Peptide-Func (graph classification) and Peptide-Struct(graph regression) . To better understand the dataset, Table 5 and Table 6 lists the statistics of 10 benchmark datasets. Table 5 provides the number of learning instances and binary tasks for each chemical dataset. Table 6 presents long-range datasets and predictive tasks for peptides. While the peptide datasets have similar numbers of learning instances, they differ in the predictive tasks.

Table 5: The Overview of Molecular Property Prediction Datasets.

| Dataset | Graphs | Avg. Nodes | Avg. Edges | Tasks |
|---------|--------|-----------|-----------|-------|
| BBBP | 2, 039 | 24.1 | 26.0 | 1 |
| Tox21 | 7, 831 | 18.6 | 19.3 | 12 |
| Toxcast | 8, 576 | 18.8 | 19.3 | 617 |
| SIDER | 1, 427 | 33.6 | 35.4 | 27 |
| Clintox | 1, 478 | 26.2 | 27.9 | 2 |
| MUV | 93, 087 | 26.4 | 28.3 | 17 |
| HIV | 41, 1127 | 25.5 | 27.5 | 1 |
| BACE | 1, 513 | 34.1 | 36.9 | 1 |

## C.2 Model and Training Configurations

In order to help better understand our proposed model and reproduce the work, we provide the parameter selection of the experiment. Table 7 provides the configurations of $G_M$, $G_F$ and $f_C$.

Table 6: The Overview of Long-Range Graph Datasets.

| Dataset | Peptides-func | Peptides-struct |
|---|---|---|
| Graphs | 15, 535 | 15, 535 |
| Avg.Nodes | 150.9 | 150.9 |
| Avg.Edges | 307.3 | 307.3 |
| Tasks | 1 | 5 |
| Classes | 10 | N/A |
| Task types | Multi-label classification | Multi-label regression |

Table 7: Model Configuration

| Models | Hyperparameters | Values |
|---|---|---|
| $9*G_M$ | Convolution type | GIN |
| | Number of atom features | 2 |
| | Number of atom features for long-range | 9 |
| | Number of edge features | 2 |
| | Number of edge features for long-range | 3 |
| | Dimension of hidden embeddings | 300 |
| | Aggregation | SUM |
| | Number of layers | 5 |
| | Readout | MEAN |
| $7*G_F$ | Convolution type | GIN |
| | Number of fragment features | 1 |
| | Dimension of hidden embeddings | 300 |
| | Aggregation | SUM |
| | Number of layers | 2 |
| | Number of layers for long-range | 5 |
| | Readout | MEAN |
| $5*f_C$ | Convolution type | Swin-B |
| | Input patches of size | 4×4 |
| | Local attention windows of size | 7×7 |
| | Input images of size | 224×224 |
| | Pretrained dataset | ImageNet-1K |

Table 8 provides the parameters of the pretraining phase. Table 9 provides the parameters of the finetuning phase. In general, our parameter selection refers to previous works Zhang et al. (2021); Luong & Singh (2024).

## D  RESULTS ON LONG-RANGE GRAPH BENCHMARK

In Table 10, we compare the MotifGrIm with GNN baselines on two long-range benchmarks, namely, PEPTIDE-FUNC (which includes 10 peptide function classification tasks) and PEPTIDE-STRUCT (which includes 5 3D structure regression tasks) Dwivedi et al. (2022). Given the substantial size of peptide fragment graphs, which hinders the efficient extraction of structural backbones, predictive learning module has been omitted from this experimental setup, i.e., $\alpha = 1.0$. The experimental results demonstrate that our proposed method significantly outperforms the other GNN baselines. Specifically, compared to the current best performing method, MotifGrIm improves by 4.1% on PEPTIDE-FUNC and 11.3% on PEPTIDE-STRUCT, whereas the performance is improved by 14.8% and 17.6%, respectively, compared to vanilla GIN. Since these tasks require recognizing long-range structural information, these improvements highlight the effectiveness of our model in capturing the global arrangement of molecules.

Table 8: Pretraining Configuration

| Benchmark | Hyperparameters | Values |
|---|---|---|
| Molecular Property Prediction | Epochs | 100 |
| | Batch size | 256 |
| | Weight decay | 1e-2 |
| | Learning rate | 1e-3 |
| | Learning rate decay | Reduce on plateau |
| | Learning rate decay factor | 1e-1 |
| | Learning rate decay patience | 5epochs |
| | weighting parameter $\alpha$ | 0.9 |
| | hybrid parameter $\beta$ | 0.7 |
| Long-range | Epochs | 100 |
| | Batch size | 256 |
| | Weight decay | 1e-2 |
| | Learning rate | 1e-3 |
| | Learning rate decay | Reduce on plateau |
| | Learning rate decay factor | 1e-1 |
| | Learning rate decay patience | 5epochs |
| | weighting parameter $\alpha$ | 1.0 |
| | hybrid parameter $\beta$ | 0.7 |

Table 9: Finetuning Configuration

| Dataset | Epoch | Batch size | Dropout | Weight decay | Learning rate | Learning rate decay | Decay steps | Decay rate |
|---|---|---|---|---|---|---|---|---|
| BBBP | 100 | 64 | 0.0 | 0.0 | 1e-4 | Step decay | 30 | 0.3 |
| Tox21 | 100 | 64 | 0.0 | 0.0 | 1e-3 | Step decay | 30 | 0.3 |
| Toxcast | 100 | 64 | 0.5 | 0.0 | 1e-3 | Step decay | 30 | 0.3 |
| SIDER | 100 | 64 | 0.5 | 0.0 | 1e-3 | Step decay | 30 | 0.3 |
| Clintox | 100 | 64 | 0.0 | 0.0 | 1e-3 | Step decay | 30 | 0.3 |
| MUV | 100 | 64 | 0.0 | 0.0 | 1e-4 | Step decay | 30 | 0.3 |
| HIV | 100 | 64 | 0.0 | 0.0 | 1e-3 | Step decay | 30 | 0.3 |
| BACE | 100 | 64 | 0.0 | 0.0 | 1e-4 | Step decay | 30 | 0.3 |
| Peptides-func | 100 | 128 | 0.0 | 0.0 | 1e-3 | Step decay | 30 | 0.5 |
| Peptides-struct | 100 | 128 | 0.0 | 0.0 | 1e-3 | Step decay | 30 | 0.5 |

Table 10: Test AP(Average Precision Score) on graph classification dataset PEPTIDE-FUNC and test MAE(Mean Absolute Error) on graph regression dataset PEPTIDE-STRUCT. These tasks involve modeling long-range interactions in large peptide molecules.

| 2*Methods | Peptide-Func Test AP | Peptide-Struct Test MAE |
|---|---|---|
| GCN | 0.5930±0.0023 | 0.3496±0.0013 |
| GCNII | 0.5543±0.0078 | 0.3471±0.0010 |
| GIN | 0.5498±0.0079 | 0.3547±0.0045 |
| GatedGCN | 0.5864±0.0077 | 0.3420±0.0013 |
| GatedGCN+RWSE | 0.6069±0.0035 | 0.3357±0.0006 |
| MotifGrIm | **0.6314±0.0049** | **0.3016±0.0008** |

Table 11: The impact of the weighting hyperparameter $\beta$ used to hybrid graph-image contrastive learning.

| Method | BBBP | Tox21 | Toxcast | SIDER | ClinTox | MUV | HIV | BACE | Avg.Rank | Avg.AUC |
|---|---|---|---|---|---|---|---|---|---|---|
| MotifGrIm($\alpha$=0.9, $\beta$=0.5) | 71.2 | 75.3 | 64.5 | **64.3** | 88.8 | 76.8 | 75.8 | 82.1 | 4.50 | 74.85 |
| MotifGrIm($\alpha$=0.9, $\beta$=0.6) | 71.6 | 75.6 | 65.3 | 63.8 | 89.5 | 78.2 | 76.9 | 83.2 | 2.88 | 75.51 |
| MotifGrIm($\alpha$=0.9, $\beta$=0.7) | 70.7 | 75.4 | **65.4** | 64.2 | **91.2** | **78.4** | **77.8** | 82.7 | **2.38** | **75.73** |
| MotifGrIm($\alpha$=0.9, $\beta$=0.8) | 71.1 | 75.2 | 65.3 | 64.1 | 89.7 | 77.6 | 76.9 | **83.7** | 3.38 | 75.45 |
| MotifGrIm($\alpha$=0.9, $\beta$=0.9) | 70.8 | 75.3 | 65.1 | 63.7 | 89.8 | 76.5 | 77.3 | 83.1 | 4.00 | 75.20 |
| MotifGrIm($\alpha$=0.9, $\beta$=1.0) | **72.6** | **75.8** | 65.2 | 63.6 | 90.5 | 75.5 | 77.3 | 82.4 | 3.38 | 75.36 |

# E    MORE PARAMETER ANALYSIS

In Section 3.4, we provided a parameter analysis of $\alpha$ and $K$. To further understand how parameter affects model performance, we conduct experiments here with different values of weighting hyperparameter $\beta$ used to hybrid graph-image contrastive learning and motif vocabulary sizes $N$.

## E.1    EFFECT OF HYBRID PARAMETER $\beta$.

In Section 2.2.3, we propose a hybrid graph-image contrastive learning method. Unlike previous multi-modal methods that only perform contrastive learning from the molecular perspective, we simultaneously leverage data from both the motif and molecular perspectives for contrastive learning between graphs and images. As shown in Equation 8, we use $\beta$ to adjust the embedding weights for the two perspectives. To analyze the effect of different weights for motif embeddings and molecular embeddings on model performance, we designed experiments to perform a parameter analysis on $\beta$. Specifically, we select $\beta$ from {0.5, 0.6, 0.7, 0.8, 0.9, 1.0} and fixed $\alpha$ at 0.9. From the experimental results show in Table 11, we can see that the performance is best when $\beta = 0.7$, clearly outperforming the case without using hybrid contrastive learning, demonstrating the effectiveness of this method. When $\beta = 0.9$, the model performance decreased compared to the case without this method. This is likely because the weight of the motif embedding was set too small, which not only failed to fully utilize the motif data but also introduced noise interference during the optimization of the molecular embedding. When $\beta = 0.5$, the model performs the worst. This may be because the motif and molecular data are treated equally, preventing the model from effectively capturing information at different levels during the optimization process.

## E.2    EFFECT OF MOTIF VOCABULARY SIZE $N$.

The size of the motif vocabulary may impact the classification performance of the model. To explore this effect, we experimented with vocabulary sizes of {200, 400, 800, 1600, 3200}. The results, summarized in Table 12, show that the best performance is achieved with a vocabulary size of 800. We hypothesize the following: when the vocabulary is too small, the method fails to capture sufficiently large motifs, leading to an overly fragmented representation of molecules and an inability to effectively extract high-level information. Conversely, a larger vocabulary leads to an increase of the number of parameters, making it more challenging to optimize the model.

Table 12: The impact of varying motif vocabulary sizes $N$.

| Method | BBBP | Tox21 | Toxcast | SIDER | ClinTox | MUV | HIV | BACE | Avg.Rank | Avg.AUC |
|---|---|---|---|---|---|---|---|---|---|---|
| MotifGrIm($size$=200) | 69.4 | 75.5 | 64.5 | **64.2** | 85.4 | 74.7 | 75.9 | 81.7 | 3.75 | 73.91 |
| MotifGrIm($size$=400) | **71.2** | **75.6** | **65.4** | 63.4 | **91.8** | 76.4 | 77.3 | 82.2 | 2.50 | 75.41 |
| MotifGrIm($size$=800) | 70.7 | 75.4 | **65.4** | **64.2** | 91.2 | **78.4** | 77.8 | 82.7 | **1.88** | **75.73** |
| MotifGrIm($size$=1600) | 70.6 | 75.4 | 64.8 | 63.4 | 84.5 | 77.4 | **78.2** | 82.9 | 3.13 | 74.61 |
| MotifGrIm($size$=3200) | 70.1 | 75.2 | 64.2 | 63.9 | 83.1 | 77.6 | 77.5 | **83.2** | 3.38 | 74.35 |

