# OpenReview forum: "MotifGrIm: Motif-Based Multi-Granularity Graph-Image Pretraining for Molecular Representation Learning"
_ICLR.cc/2026/Conference — Submitted to ICLR 2026_

### Official Review · Reviewer_zGWg · 2025-10-18

**Soundness:** 3
**Presentation:** 3
**Contribution:** 2
**Rating:** 4
**Confidence:** 4

**Summary:**

The authors present MotifGrIm, a motif-based multi-granularity pretraining framework for molecular representation learning that jointly leverages graph and image modalities, integrating motif-level information into both domains: representing motifs as structured subgraphs in the graph encoder and as distinct background patterns in the image encoder.
A multi-granularity prediction strategy and contrastive learning are proposed to capture molecular information at different hierarchical levels (atom, motif, and molecule). MotifGrIm also aims to mitigate two core issues: Over-smoothing and the loss of motif connectivity information

**Strengths:**

- The idea about the integration of motif information into the image domain is conceptually new, bridging structural and visual molecular representations.

- The use of motifs as intermediate granularity between atom-level and molecule-level features effectively captures hierarchical molecular semantics.

**Weaknesses:**

- The rationale for involving molecular images is not fully convincing. It remains unclear why image-domain motif augmentation leads to better representations or whether it simply introduces auxiliary regularization.

- The framework’s expressive power relative to Weisfeiler–Lehman (WL) or motif-aware GNNs is not discussed.

**Questions:**

- What is the intuition behind incorporating motifs into molecular images? Do motif-based backgrounds actually enhance structural alignment, or could they introduce artifacts?

- Does the motif-based message passing or alignment extend the expressive power beyond 1-WL equivalence?

- Although motif integration offers interpretability potential, the paper does not provide visual or quantitative evidence linking learned features to chemically meaningful motifs.

---

### Official Review · Reviewer_9X1e · 2025-10-21

**Soundness:** 2
**Presentation:** 3
**Contribution:** 2
**Rating:** 2
**Confidence:** 5

**Summary:**

The paper proposes MotifGrIm, a motif-based, multi-granularity graph-image pretraining framework for molecular representation learning. It constructs four synchronized views per molecule (molecular graph, motif graph, molecular image, motif image) and performs contrastive learning within and across modalities.

**Strengths:**

The idea of injecting motif knowledge into an image view and aligning it with graphs is interesting.

**Weaknesses:**

1.	The paper repeatedly claims “we incorporate motifs into the image domain for the first time”. However, MaskMol [1] (Cheng et al., 2024; BMC Biology, 2025) already performs knowledge-guided molecular image pre-training with pixel masking that explicitly incorporates atomic, bond, and motif knowledge in the image view. Therefore, I suggest that the author give an accurate description of the innovation.
2.	The model is pretrained on only ~456K molecules, which is far smaller than commonly used corpora in recent baselines (e.g., CGIP ≈ 10M, AttrMasking ≈ 2M). As a result, the head-to-head comparisons are not on equal footing. I suggest that the author use a larger pre-training dataset for training, on the one hand, to demonstrate the effectiveness of the pre-training method in parallel comparison with other baselines, and on the other hand, to demonstrate the scalability of the model.
3.	The method encodes motifs via background colors, while standard 2D molecular images already use colors for atom types (e.g., O=red, N=blue, S=yellow) and lines for bonds. In such sparse, line-art images, colored motif backgrounds can collide with or wash out atom-type colors, potentially creating ambiguous or misleading cues for the CNN (e.g., a red oxygen symbol over a reddish motif background). The paper does not specify how these collisions are avoided or constrained.
4.	The paper uses graph + image pretraining but does not compare against strong image-only SSL methods (such as VideoMol [2], MaskMol [1], and so on), leaving the benefit of multi-modal unclear.

[1] MaskMol: knowledge-guided molecular image pre-training framework for activity cliffs with pixel masking, BMC Biology, 2025.
[2] A molecular video-derived foundation model for scientific drug discovery, Nature Communications, 2024.

**Questions:**

Consistent with Weakness

---

### Official Review · Reviewer_myBA · 2025-10-30

**Soundness:** 2
**Presentation:** 2
**Contribution:** 2
**Rating:** 2
**Confidence:** 4

**Summary:**

This paper presents MotifGrIm, a motif-based multi-granularity graph-image pretraining framework for molecular representation learning. It incorporates motif information into the image modality. The framework generates four views of each molecule—molecular graph, motif graph, molecular image, and motif image—and employs contrastive learning within and across modalities, along with multi-granularity prediction tasks, to capture both local substructures and global topology. Experiments demonstrate state-of-the-art performance on eight molecular property prediction datasets.

**Strengths:**

This paper is well-written and is easy to understand.

**Weaknesses:**

1.  Limited Gain from Graph-Image Multimodal Pretraining: The paper claims that graph-image multimodal pretraining enhances molecular representation by combining structural and visual information. However, this approach may offer diminishing returns compared to other modalities. Specifically, molecular images largely encapsulate information already present in graph structures (e.g., atoms and bonds depicted in 2D layouts), meaning the additional gains from images could be marginal. In contrast, textual descriptions (e.g., from scientific literature) provide rich, high-level semantic information—such as functional properties or biological contexts—that graphs alone cannot capture.

2. Questionable Contrastive Learning Strategy for Motif Alignment: The paper employs contrastive learning to align representations between molecular graphs and motif graphs, as well as between molecular images and motif images. This strategy aims to pull these views closer in the embedding space, but it raises conceptual concerns. For instance, aligning a molecule's global representation with its substructures (motifs) could lead to semantic inconsistencies: different motifs within the same molecule might have divergent structures or functions (e.g., a hydrophobic group versus a polar group), yet they are forced to be proximate in the feature space simply because they belong to the same molecule.

3. Outdated Baselines Weaken SOTA Claims: The experimental evaluation compares MotifGrIm against several existing methods, but many baselines originate from works published in 2023 or earlier. This undermines the paper's claim of state-of-the-art (SOTA) performance. The authors should compare with more recent baselines such as [1].

[1] Advancing Molecular Graph-Text Pre-training via Fine-grained Alignment

**Questions:**

1. Could the authors elaborate on the underlying hypothesis for minimizing the distance between a whole-molecule embedding and its motif-level embeddings in the latent space?

2. Could the authors provide results or discuss potential downstream tasks that specifically leverage the image encoder?

---

### Official Review · Reviewer_dFht · 2025-10-31

**Soundness:** 2
**Presentation:** 2
**Contribution:** 1
**Rating:** 2
**Confidence:** 5

**Summary:**

This paper introduces MotifGrIm, a pre-training framework for molecular representation learning that operates on multiple granularities (molecule and motif) and modalities (graph and image). The core contribution is the introduction of motif-level information into the image domain by generating "motif images," where different molecular motifs are highlighted with distinct colors. The framework employs contrastive and predictive learning objectives to align representations within and across these different views. The authors demonstrate the effectiveness of MotifGrIm through experiments on molecular property prediction and long-range interaction benchmarks.

**Strengths:**

1. The framework systematically integrates several existing learning paradigms, including multi-modal contrastive learning (graph-image), multi-granularity learning (molecule-motif), and auxiliary predictive tasks, into a single, unified pre-training process.

2. The paper is evaluated on a wide array of standard downstream tasks, including eight MoleculeNet benchmarks and two long-range benchmarks. The inclusion of extensive ablation studies provides some insight into the utility of the framework's different components.

**Weaknesses:**

1. Limited Novelty and Incremental Contribution

The primary weakness of this work lies in its limited conceptual novelty. The core ideas—leveraging motifs for representation learning and using graph-image multi-modal pre-training—have been individually explored in prior work. The main contribution of this paper is to combine these two known concepts. The specific implementation of coloring motifs in an image, while an interesting detail, feels more like an incremental engineering step than a fundamental advance in the field. Consequently, the work reads more like a skillful assembly of existing components rather than a novel method with a strong, original foundation.

2. High Framework Complexity for Marginal Gain

The proposed MotifGrIm framework is exceedingly complex. It involves generating four distinct data views, training three separate encoders (two GNNs, one Vision Transformer), and optimizing a complex objective function with multiple contrastive and predictive losses. Despite this high complexity, the reported performance gains over the strongest baseline (MoleculeSTM) are marginal (an average ROC-AUC improvement of 1.16%). The paper fails to justify why such a complex approach is warranted for a modest improvement, raising questions about its practical value and elegance.

3. Insufficient Ablation to Justify the Core Idea

The central claimed novelty is the use of "motif images" to make the image encoder substructure-aware. However, the ablation studies are insufficient to validate this specific contribution. The experiments only compare against variants that remove the entire image modality (w.o. IH and w.o. IPH). A critical and missing baseline would be a model variant that uses standard, uncolored molecular images for the image-based contrastive learning tasks. Without this comparison, it is impossible to determine whether the performance gain comes from simply adding an image modality or specifically from the proposed motif-coloring scheme.

4. Omission of Computational Cost Analysis

Training a large vision transformer on image data is substantially more resource-intensive than graph-only pre-training methods. A detailed analysis of pre-training time and memory consumption is necessary to assess the practicality and scalability of the method. The lack of this analysis makes it difficult to evaluate the method's efficiency.

5. Uncertainty Regarding Generalizability

The framework's performance is tied to a specific "Principal Subgraph Mining" algorithm for motif extraction. The paper does not investigate how robust the model is to different motif-finding algorithms (e.g., chemistry-based fragmentation rules like BRICS). This dependency on a single, heuristic-based algorithm raises concerns about the generalizability and robustness of the entire approach.

**Questions:**

1. Given that motif-based pre-training and graph-image contrastive learning have been explored in separate prior works, could you clarify the primary conceptual novelty of your framework beyond their direct combination? Furthermore, please justify the trade-off between the significant complexity of your proposed framework and the marginal performance gains observed over strong baselines.

2. Your central technical contribution appears to be the motif-coloring scheme. To properly validate its effectiveness, it is essential to compare it against a model that also uses the image modality but without any motif coloring (i.e., using standard molecular images). Can you provide results for this crucial ablation or explain why it was omitted?

3. Could you provide a detailed analysis of the pre-training computational cost (e.g., training time, GPU memory) of MotifGrIm and compare it to both a strong graph-only baseline (like MGSSL) and a strong multi-modal baseline (like MoleculeSTM)? This information is critical for understanding the practical implications of your method.

---

### Meta-Review · Area_Chair_zaSa · 2026-01-06

**Summary:**

Reviewers share negative sentiment of this paper. In particular, reviewers are mostly concerned about the following key issues:

- Limited novelty, as prior work on molecular image pre-training already incorporated motif knowledge

- Gain is marginal from graph-image multimodal pre-training

- More recent baselines should be used, especially strong image-only methods.

In my opinion, these issues are crucial and significant, and must be addressed before publication. However, the authors did not provide any response. Hence, rejection is recommended.

**Reviewer Concerns:**

Authors did not provide rebuttal.

**Reviewer Scores:**

Authors did not provide rebuttal.

---

### Decision · Program_Chairs · 2026-01-26

Reject